# Falls in Parkinson’s Disease Subtypes: Risk Factors, Locations and Circumstances

**DOI:** 10.3390/ijerph16122216

**Published:** 2019-06-23

**Authors:** Paulo H. S. Pelicioni, Jasmine C. Menant, Mark D. Latt, Stephen R. Lord

**Affiliations:** 1Neuroscience Research Australia, NSchool of Public Health and Community Medicine, University of New South Wales, Sydney, NSW 2052, Australia; p.pelicioni@neura.edu.au (P.H.S.P.); j.menant@neura.edu.au (J.C.M.); 2Department of Aged Care, Royal Prince Alfred Hospital, Sydney, NSW 2050, Australia; mark.latt@sydney.edu.au

**Keywords:** Parkinson’s disease, accidental falls, subtypes, postural balance, executive function, freezing of gait

## Abstract

People with Parkinson’s disease (PD) can be classified into those with postural instability and gait difficulty (PIGD subtype) and those manifesting tremor as the main symptoms (non-PIGD subtype). In a prospective cohort study of 113 people with PD we aimed to contrast fall rates and circumstances as well as a range of disease-related, clinical, and functional measures between the PD subtypes. Compared with non-PIGD participants, PIGD participants were significantly more likely to suffer more falls overall as well as more falls due to freezing of gait, balance-related falls and falls at home. The PIGD group also performed significantly worse in a range of fall-related clinical and functional measures including general cognitive status, executive function, quadriceps muscle strength, postural sway and the timed up and go test. These findings document the extent to which people with the PIGD subtype are at increased risk of falls, the circumstances in which they fall and their disease-related, clinical and functional impairments.

## 1. Introduction

Falls are a significant cause of disability, lost independence and reduced quality of life in people with Parkinson’s disease (PD) [1,2]. Prospective studies show that between 45% and 68% of people with PD will fall each year [3,4], with a large proportion (50–86%) falling recurrently [5]. In addition, their risk of falls and fractures rise steadily from 40 years of age, much earlier than in healthy individuals [6]. The consequences of falls are devastating and include restriction of activities of daily living, fear of falling, high levels of caregiver stress and injuries [1]. In fact, the incidence of hip fracture is four times that for older people of the same age without PD [7]. This has significant economic consequences as the costs of fall-related fractures in people with PD are close to double those in healthy older people [2].

Many risk factors for falls in PD have been identified. These include freezing of gait (FOG), cognitive impairment, poor leaning balance, previous falls, lower limb weakness and slow gait speed [4,8,9]. In addition, people with PD with the postural instability and gait difficulty (PIGD) subtype have also been identified as having an increased risk of falls. This subtype has a predominance of postural instability and gait impairment as opposed to the tremor dominant (TD) subtype, for which there is a predominance of resting and postural tremor [10]. Only a few studies, however, have investigated whether people with the PIGD subtype fall more frequently [11,12,13], and such work is based solely on retrospective surveys, limited by differential to recall bias.

The circumstances in which falls occur may provide insights into underlying causes of falls and possible fall prevention strategies. For example, in people with PD, falls that occur outdoors are primarily due to slips and trips, whereas falls that occur indoors are related more strongly to lower limb weakness, vertigo and postural instability [14]. However, no studies to date have examined differences between the PIGD and TD subtypes in the circumstances of falls, including fall locations. In addition, only a few studies have contrasted cognitive, functional and mobility measures between the two PD subtypes [10,11,15]. Such between-group comparisons are important to identify key impairments that might be amenable to intervention and hence guide the prescription of evidence-based treatments in clinical practice.

Therefore, the aims of this study were to determine differences between the PIGD and non-PIGD PD subtypes in: (1) incidence of falls in a prospective study design, (2) fall locations and fall types, relating to the occurrence of FOG, balance impairment and syncope and (3) a range of disease-related, cognitive, functional and mobility measures. We hypothesised that compared with the non-PIGD group, the PIGD subtype would suffer more falls due to syncope, instability and FOG; more falls at home and more falls overall; and exhibit cognitive, functional and mobility impairments on clinical assessment.

## 2. Materials and Methods

### 2.1. Participants

This study comprised a secondary analysis from a prospective study of falls [8]. One hundred and thirteen people with idiopathic PD diagnosed according to the UK PD Brain Bank criteria [16] were recruited from within the city of Sydney, NSW. Volunteers were recruited from the following sources: 36 (31%) from a hospital outpatient clinic, 15 (13%) from a volunteer database, 20 (18%) from PD support groups and 42 (37%) from the general community. The exclusion criteria comprised: An inability to walk unassisted without a walking aid during the tests, a Mini-Mental State Examination (MMSE) [17] score <24, an atypical Parkinsonian syndrome evidence of psychosis, neuroleptic use, vertigo, epilepsy, stroke, transient ischaemic attacks, syncope, uncompensated heart failure or valvular heart disease. Participants were assessed in their typical “on” phase of the levodopa treatment cycle. The protocol was approved by the Human Studies Ethics Committee at the University of Sydney (approval number: HREC 2002/3/4.18 (1401)) and informed consent was obtained from all participants prior to their participation.

### 2.2. Assessments

#### 2.2.1. Subtype Classification

The mean tremor score was calculated as the mean of Unified Parkinson’s Disease Rating Scale (UPDRS) part II, item 16 (tremor) and UPDRS part III, items 20 (rest tremor) and 21 (action tremor) scores. The mean PIGD score was calculated as the mean of UPDRS part II, items 13 (falling), 14 (freezing) and 15 (walking) and UPDRS III, items 29 (gait) and 30 (postural stability) scores [10]. The ratio of mean tremor score to mean PIGD score was then calculated to determine the PD subtype: ratios ≥1.5 identified participants with the TD subtype (*n* = 13), ratio scores ≤1.0 the PIGD subtype (*n* = 67) and ratios between 1.01 and 1.49 the indeterminate subtype (*n* = 2). Due to relatively small sample numbers, the TD and indeterminate groups were combined to form a single non-PIGD group (*n* = 46) for all analyses [10].

#### 2.2.2. PD-Related and Health Measures

In addition to the above PD subtype classification, several other PD related measures were collected. These included duration of disease since first symptoms; stage of the disease according to the Hoehn and Yahr scale [18]; Presence of Rigidity, Axial Posture, Bradykinesia and Dyskinesia according to the UPDRS items; UPDRS part I, II, II, IV and total scores [19]; levodopa daily dosage, dopamine agonist, anticholinergic medication and Catechol-O-Methyl Transferase (COMT) inhibitor use. In addition, information on falls in the past year, walking aid use inside and outside the home and non-PD medication use was recorded.

#### 2.2.3. Cognition

The MMSE was used to assess global cognition [17] and the Frontal Assessment Battery (FAB) was used to assess executive functioning [20].

#### 2.2.4. Sensorimotor, Balance, Gait and Mobility

Participants were assessed using the Physiological Profile Assessment (PPA) [21] which comprises tests evaluating key functions of the human balance system: Peripheral sensation, visual contrast sensitivity, lower limb strength, simple reaction time and postural sway when standing on a compliant surface. An individual composite fall risk score was computed from an algorithm including the scores achieved in each test. This physiological fall risk assessment has been shown to predict recurrent falls in community-dwelling older people with an accuracy of 75% [22]. The coordinated stability test was used to assess controlled leaning balance, i.e. how participants adjust their balance in a steady and coordinated manner when near the limits of their base of support. Higher scores indicate poorer dynamic postural stability [23].

Gait analysis was performed using a tri-axial piezo-resistant accelerometer attached to the participant’s pelvis on a belt at the level of the sacrum. Participants performed one walking trial at self-selected speed along a 20 m corridor and data collected from the middle 15 m of steady state walking were analysed. Average gait speed was calculated by dividing the walking distance by the total time taken to complete the distance. Step time variability was computed from the extracted acceleration data (standard deviations between successive heel contacts over the middle 15 m of steady state walking) [15]. Step time variability was calculated from the average of 32 steps (range 14–204).

Functional mobility was assessed with the timed up and go (TUG) test [24]. Participants were asked to rise from a chair, walk forward three meters at their usual walking pace, turn 180 degrees, walk back to the chair and sit down.

#### 2.2.5. Orthostatic Hypotension

Orthostatic hypotension was defined as a fall in systolic blood pressure by 20 mmHg or more and/or in diastolic blood pressure by 10 mmHg or more, recorded with a sphygmomanometer on the left arm, during the first 3 min of standing up from sitting [25].

#### 2.2.6. Falls

Falls were defined as unexpected events which resulted in the participant unintentionally coming to the ground, floor or other lower level [26]. Falls were collected prospectively for 12 months using monthly calendars. All participants who reported a fall were telephoned by a single experienced geriatrician (M.D.L.) to verify the falls circumstances and any related injuries [26]. We classified falls within three types: falls that occurred immediately following FOG (FOG falls); Falls resulting from a slip, trip or loss of balance (balance-related falls) [27]; and falls due to possible syncope or pre-syncope (syncopal and other falls) [27]. Falls that occurred inside the participant’s house or immediate surroundings (outside stairs, garage, and garden) were classified as at home falls, with the remainder classified as away from home falls.

### 2.3. Statistical Analysis

Continuous data were inspected for skewed distributions, and log-transformed if required. Differences between PIGD and non-PIGD groups with respect to demographics, physical, disease-related, clinical and functional variables were examined with Student *t*-tests for independent samples (normally distributed data), Mann–Whitney *U*-tests (non-parametric data) or chi square tests (categorical data). These data were also compared between the PIGD and non-PIGD groups adjusting for disease duration: analysis of covariance (continuous measures) or Mantel–Haenzel adjustments (categorical data). For the Mantel–Haenzel adjustments, disease duration was categorized as short (≤5 years, *n* = 33.6%), medium (6–11, *n* = 35.4%) and long durations (≥12 years, *n* = 31%). Finally, the relationship between PD subtype groups and fall outcomes were examined using the incidence rate ratios from negative binomial regression adjusting by disease duration. Geometric means of fall rates for the two groups are also presented. Geometric means provide a measure of central tendency that uses the product of the values of a variable (as opposed to the arithmetic mean which uses their sum) to accommodate for right-skewed distributions [28]. For these calculations, 0.5 was added to numerators so data for participants with no falls during the 12-months follow-up could be included. Data were analysed using IBM SPSS v. 25 for Windows (SPSS, Inc., Chicago, IL) and significance levels were set at 0.05.

## 3. Results

### 3.1. Parkinson’s Disease and Health-Related Characteristics of the PIGD and Non-PIGD Participants

Table 1 presents the demographic, disease-related and clinical characteristics data of the PIGD and non-PIGD participants. No between-group differences were found for age, sex, height or body mass. In unadjusted analyses, the participants with PIGD had longer disease durations, higher UPDRS parts I, II, IV and total scores, and more likely to have more advanced Hoehn and Yahr stages, worse leg/axial rigidity, worse dyskinesia, FOG and higher levodopa dosage intake. Participants with PIGD were also more likely to be taking anticholinergic medications and report previous falls and the use of walking aids inside and outside of their homes. With the exception of UPDRS part IV scores, dyskinesia and anticholinergic medication use, these associations remained statistically significant when adjusting for Parkinson’s disease duration.

### 3.2. Cognitive, Sensorimotor, Balance, Gait, Mobility and Cardiovascular Measures: PD Subtype Comparisons

Table 2 presents the cognitive, sensorimotor, balance, gait, mobility and cardiovascular measures for the PIGD and non-PIGD participants. In unadjusted analyses, the participants with PIGD showed lower MMSE and FAB scores, reduced quadriceps strength, greater postural sway, higher PPA scores, worse coordinated leaning balance as assessed with the coordinated stability test, and longer TUG times. With the exception of the PPA and coordinated leaning balance measures, these associations remained statistically significant when adjusting for PD duration.

### 3.3. Falls

A total of 2043 falls were reported. Of these, 124 falls (6%) were reported by the non-PIGD and 1919 (94%) were reported by the PIGD group. In terms of fall types, 1249 were FOG related, 537 were balance related and 257 were syncopal and other falls. Falls related outcome adjusted for disease duration for the non-PIGD and PIGD groups are presented in Table 3. Compared with the non-PIGD participants, more PIGD participants suffered falls in general as well as more FOG-related, balance-related and at-home falls. These findings were also mirrored in the fall rate analyses.

## 4. Discussion

In this prospective cohort study of 113 people with PD, we contrasted fall rates and circumstances as well as a range of disease-related, clinical and functional measures between PD motor subtypes. We found that compared with non-PIGD participants, PIGD participants were significantly more likely to suffer more falls overall as well as more falls due to FOG, balance-related falls and falls at home. After adjusting for disease duration, the PIGD group also performed significantly worse in a range of fall-related clinical and functional measures including general cognitive status, executive function, quadriceps muscle strength, postural sway and TUG mobility. These findings are consistent with our hypotheses and document the extent to which people with PD with the PIGD are at elevated risk of falls.

It has been suggested the TD-PIGD subtype distinction may reflect different stages of PD rather than different disorders [29], and as anticipated, the PIGD group had longer disease durations than the non-PIGD group in our cohort. Accordingly, we adjusted for PD duration in our subtype comparisons and some measures (motor complications assessed by UPDRS part IV, dyskinesia, anticholinergic medication, risk of falls assessed by the PPA and controlled leaning balance) did not remain statistically significant. However, many disease-related, cognitive, sensorimotor, balance, gait and mobility measures did remain significant discriminators of PD subtype after adjustment for disease duration. Therefore, it appears the PD subtype classification, as used in this study, is useful for identifying people with PD at increased fall risk as well as elucidating possible underlying causes of falls.

Previous work addressing fall risk in the PIGD subtypes has used retrospective study designs [11,12,13]. Retrospective designs are not only limited by recall bias, but also by circular comparisons, in that past falls comprise a component of the PIGD subtype classification. In contrast, our prospective findings are free of these limitations and document the greatly increased risk of falls prospectively. Our findings reveal that compared with the non-PIGD group, the PIGD group have a 170 times greater rate of FOG-related falls, a 5 times greater rate of balance-related falls, a 10 times greater rate of at-home falls and a 10 times greater rate of falls overall. In contrast to FOG-related and balance-related falls, the non-PIGD and PIGD groups did not differ with respect to either syncopal and other falls (often associated with pre-syncope or syncope events [27]) or the presence of orthostatic hypotension. This suggests that such cardiovascular related factors do not contribute to the elevated fall risk evident in the PIGD group.

In parallel to higher fall rates, the PIGD group had a greater prevalence of disease-related, cognitive, sensorimotor, balance, gait and mobility factors known to increase fall risk in people with PD [4,8,30]. While some of these factors comprise measures of disease severity, many help explain why many PIGD subtype participants fall frequently. Our findings of an elevated prevalence of FOG-related falls in the PIGD group complement previous studies [8,9,30] that have shown FOG is a strong risk factor for falls. Furthermore, the reduced quadriceps strength, poor balance and reduced mobility (slower TUG test performance) exhibited by the PIGD group might explain their increased prevalence of balance-related falls.

Bloem et al. [3] have reported that people with PD fall more at home, while healthy older people fall more outside; the former being due to disease-related mobility impairments and the latter due to greater exposure to unexpected hazards and circumstances. Our study builds on this work by showing that those with the PIGD subtype fall significantly more at home than the non-PIGD subtype. In addition to their multiple cognitive and physical impairments, fall risk in the PIGD group may be exacerbated by walking in the more confined space of the home in situations that can trigger FOG such as gait initiation, short walks and turns [31] and subsequent falls. The more similar fall rates between the PD subtypes for falls away from home may reflect those with impaired balance and mobility spending less time away from their homes with resultant limited exposure to this more hazardous environment. Future studies should measure the number of steps of the amount of physical activity required before a person falls to examine this question further.

Strengths of the study include: the inclusion of a diverse range of putative risk factors and the prospective ascertainment of falls over a 12-month period. We also acknowledge certain limitations. First, we had a sample chosen by convenience and due to the relatively small sample the TD and Intermediate subtypes were combined into one non-PIGD group. Second, we acknowledge that despite using the gold standard method of ascertaining monthly falls data via postal calendars and ensuing telephone calls, a recall bias regarding the circumstances of the falls is possible and could have led to some misclassifications of the fall types reported. Third, fall location information was obtained for approximately only half of the reported falls. This was due to the difficulty in reporting such information for those who suffered very frequent falls. Finally, the single centre nature of the study conducted in a metropolitan area may limit the generalizability of the study, even though participants were community-dwellers who were drawn from a range of sources: Hospital outpatients department and support groups. Future studies could include large samples, as well as participants from multiple sites and geographical areas to allow greater generalizability of the study findings. Larger participant numbers in each PD subtype would also allow the investigation of risk factors for falls within each subtype.

Our findings have important clinical implications in that they document clinical, medical and sensorimotor impairments in people with the PD PIGD subtype, some of which may be amenable to intervention. Future studies could address one or more of these risk factors relating to executive dysfunction, FOG, quadriceps weakness, postural instability and poor mobility in randomized controlled trials. Additionally, given the high prevalence of at home falls in the PIGD group, occupational therapy interventions based on safe mobility training and removal of environmental hazards may help prevent falls in the home setting.

## 5. Conclusions

The study findings document the extent to which people with the PIGD subtype are at increased risk of falls, the circumstances in which they fall and their disease-related, clinical and functional impairments. Compared with non-PIGD participants, PIGD participants were significantly more likely to suffer falls, more falls overall, as well as more falls due to FOG, balance-related falls and falls at home. The PIGD group also performed significantly worse in a range of fall-related clinical and functional measures including general cognitive status, executive function, quadriceps muscle strength, postural sway and the TUG test. This information may prove useful for informing cognitive, physical and environmental interventions to prevent falls in this high-risk group.

## Figures and Tables

**Table 1 ijerph-16-02216-t001:** Parkinson’s disease and health-related characteristics for the non-postural instability and gait difficulty (PIGD) and PIGD groups. Data are presented as mean (SD) unless stated otherwise.

	Non-PIGD	PIGD	Unadjusted	Adjusted #
	(*n* = 46)	(*n* = 67)	*p*	*p*
Demographic				
Sex (% Men)	26 (56)	38 (56)	0.984	-
Age (years)	66.3 (9.7)	66.1 (9.5)	0.917	-
Height (cm)	171 (8)	171 (9)	0.781	-
Body Mass (kg)	72 (12)	73 (19)	0.938	-
PD-related				
Disease duration (years)	7.2 (6.3)	10.3 (6.1)	0.024	-
UPDRS part I (score)	1.5 (1.5)	2.7 (1.8)	<0.001	0.008
UPDRS part II (score)	5.2 (3.6)	11.3 (6.2)	<0.001	<0.001
UPDRS part III (score)	14.9 (9.8)	17.1 (9.4)	0.161	0.28
UPDRS part IV (score)	2.8 (2.6)	3.8 (2.8)	0.035	0.291
UPDRS total (score)	26.0 (13.8)	36.7 (15.5)	<0.001	0.001
TD (score)	2.8 (2.0)	0.7 (1.0)	<0.001	<0.001
PIGD (score)	1.1 (1.1)	4.7 (3.0)	<0.001	<0.001
Hoehn & Yahr stages			<0.001	<0.001
Stage I (%)	25 (54)	14 (21)		
Stage II (%)	14 (30)	19 (28)		
Stage III (%)	6 (13)	27 (40)		
Stage IV (%)	1 (2)	7 (10)		
Rigidity (%) ^	40 (87)	41 (61)	0.003	0.012
Stooped Posture (%) ^	29 (63)	53 (79)	0.085	0.094
Bradykinesia (%) ^	36 (78)	57 (85)	0.453	0.653
Dyskinesia (%) ^	8 (17)	24 (36)	0.036	0.375
Levodopa dosage daily intake (%)				
<750 mg	40 (87)	35 (52)	<0.001	0.003
>750 mg	6 (13)	32 (48)		
FOG (%)	8 (17)	41 (61)	<0.001	<0.001
Anticholinergic medication (%)	0	8 (12)	0.020	0.105
Dopamine agonist medication (%)	9 (20)	10 (15)	0.611	0.698
COMT inhibitor medication (%)	3 (6)	10 (15)	0.234	0.422
General health-related				
Past fallers (%)	15 (33)	46 (69)	<0.001	0.001
Walking aid used—inside (%)	3 (6)	21 (31)	0.002	0.017
Walking aid used—outside (%)	5 (11)	27 (40)	0.001	0.006
Drugs daily intake >5 (%)	16 (35)	33 (49)	0.176	0.473

# Adjusted for disease duration. PD—Parkinson’s disease; UPDRS—Unified Parkinson’s Disease Rating Scale; TD: tremor dominant; PIGD: Postural Instability and Gait Difficulty; FOG—Freezing of gait; COMT: Catechol-O-Methyl Transferase. ^ noted as part of the UPDRS assessment.

**Table 2 ijerph-16-02216-t002:** Cognitive, sensorimotor, balance, gait, mobility and cardiovascular characteristics for the non-PIGD and PIGD groups. Data are presented as mean (SD) unless stated otherwise.

	Non-PIGD	PIGD	Unadjusted	Adjusted #
	(*n* = 46)	(*n* = 67)	*p*	*p*
MMSE (score)	29.6 (1.0)	28.7 (2.5)	0.004	0.046
FAB (score)	17.1 (1.7)	15.1 (3.7)	0.001	0.002
Contrast Sensitivity (dB)	19.6 (2.1)	19.8 (1.8)	0.695	0.459
Proprioception (degrees of error)	2.3 (1.1)	2.5 (1.5)	0.424	0.455
Quadriceps strength (kg)	24.7 (10.7)	20.2 (8.6)	0.014	0.016
Hand reaction time (ms)	281 (73)	300 (109)	0.265	0.475
Postural sway (mm)	159 (149)	265 (237)	0.008	0.047
PPA (score)	0.95 (1.29)	1.58 (1.54)	0.021	0.156
Coordinated stability (score)	12.9 (14.1)	18.2 (11.8)	0.039	0.209
Step velocity (m/s)	1.05 (0.19)	0.95 (0.34)	0.097	0.214
Step time variability (ms)	67 (29)	79 (65)	0.491	0.611
TUG (s)	8.3 (2.6)	12.5 (8.4)	<0.001	<0.001
Orthostatic hypotension (%)	6 (13)	10 (15)	1	0.922

# Adjusted for disease duration. MMSE—Mini-Mental State Examination; FAB—Frontal Assessment Battery; PPA—Physiological Profile Assessment; TUG—Timed up and go test. Note: higher scores indicate better performances in the MMSE, FAB, contrast sensitivity, quadriceps strength, step velocity and worse performances in the proprioception, hand reaction time, postural sway, PPA, coordinated stability, step time variability and TUG.

**Table 3 ijerph-16-02216-t003:** Fall outcomes in the PD subtypes.

		Non-PIGD	PIGD	Subtype Comparison
		(*n* = 46)	(*n* = 67)	
		*n* (%)	*n* (%)	IRR (95% confidence interval)
Fallers	≥1 fall	9 (20)	37 (55)	3.04 (1.46–6.34)
	≥1 FOG-related fall	2 (4)	23 (34)	6.76 (1.58–28.91)
	≥1 balance-related fall	7 (15)	26 (39)	2.50 (1.07–5.86)
	≥1 syncopal and other fall	1 (2)	7 (10)	5.03 (0.60–42.21)
	≥1 at home falls ^#^	5 (11)	30 (45)	4.10 (1.41–11.91)
	≥1 away from home falls ^#^	4 (9)	10 (15)	3.25 (1.08–9.76)
		GM ^‡^	GM ^‡^	IRR (95% confidence interval)
Falls	All fall types	0.83	3.48	10.21 (3.22–32.44)
	FOG related falls	0.54	1.62	156.30 (23.89–1022.45)
	Balance related falls	0.74	1.30	4.72 (1.33–16.70)
	Syncopal and other falls	0.55	0.67	1.58 (0.14–17.41)
	At home falls ^#^	0.66	1.76	9.94 (2.50–39.61)
	Away from home falls ^#^	0.62	0.86	1.32 (0.98–17.87)

FOG—Freezing of gait; IRR: Incidence Rate Ratio. ^#^ Fall location data from 1014 falls. ^‡^ Geometric mean of individual fall rates with 0.5 added to all numerators to enable the inclusion of participants with zero falls during the follow-up period in the comparisons.

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
