# Peer review of "Falls in Parkinson’s Disease Subtypes: Risk Factors, Locations and Circumstances"

_ijerph, 2019, doi:10.3390/ijerph16122216_

Reviewer 1 Report

In this article Authors describe the risk factors and the context in which patients with different subtypes of Parkinson's Disease fall more frequently.

This study is well designed and addresses a critical topic for Movement Disorders neurologists, both clinically and scientifically.

The main issue to be highlighted is also discussed by the authors: PIGD patients have a significantly longer duration of illness. This element can reflect different stages of PD rather than different subtypes. Accordingly, Authors adjusted for disease duration comparisons and measurements.

I suggest to better clarify in paragraph 2.2.6 Falls the way in which the details of the falls were verified by telephone. Falls have been classified into three types (FOG, balance-related and syncopal) and often, in clinical practice, it is difficult to understand the patient's explanations for making a correct anamnesis of these episodes. Has a structured questionnaire been used for all patients? Have any margins of error been identified? I also suggest that the authors debate this issue in the Discussion.

A last observation for the benefit of the inexperienced clinician in statistical analysis: I advice to the authors to better understand the meaning of the results of table 3: geometric mean of individual fall rates. It would also be interesting to know if patients with a large number of falls and their clinical characteristics have been identified.

Author Response

Comments: In this article Authors describe the risk factors and the context in which patients with different subtypes of Parkinson's Disease fall more frequently. 

This study is well designed and addresses a critical topic for Movement Disorders neurologists, both clinically and scientifically.

The main issue to be highlighted is also discussed by the authors: PIGD patients have a significantly longer duration of illness. This element can reflect different stages of PD rather than different subtypes. Accordingly, Authors adjusted for disease duration comparisons and measurements.

I suggest to better clarify in paragraph 2.2.6 Falls the way in which the details of the falls were verified by telephone. Falls have been classified into three types (FOG, balance-related and syncopal) and often, in clinical practice, it is difficult to understand the patient's explanations for making a correct anamnesis of these episodes. Has a structured questionnaire been used for all patients? Have any margins of error been identified? I also suggest that the authors debate this issue in the Discussion.

Our response: Participants who reported falls were contacted by telephone by a single experienced geriatrician (ML) to verify the circumstances of the fall(s) and if they resulted in any injuries. We have now added this information to section 2.2.6 Falls: “All participants who reported a fall were telephoned by a single experienced geriatrician (ML) to verify the falls circumstances and any related injuries”. In the limitations section of the discussion, we now acknowledge that a recall bias is still possible despite the telephone call being made within the same month as that of reporting the fall: “Second, we acknowledge that despite using the gold standard method of ascertaining monthly falls data via postal calendars and ensuing telephone calls, a recall bias regarding the circumstances of the falls is possible and could have led to some misclassifications of the fall types reported”. 

Reviewer 2 Report

This reader noticed on the demographic table only males are presented (Non-PIGD N=46 and PIGD N=67).  Did the study not include women?  It will be helpful to the reader to see this mentioned upfront as well as in the limitation of the study.  

Also, were the participants excluded if they had a history of chronic illness? (diabetes, arthritis, hx of stroke etc?).

Author Response

Comments: This reader noticed on the demographic table only males are presented (Non-PIGD N=46 and PIGD N=67).  Did the study not include women?  It will be helpful to the reader to see this mentioned upfront as well as in the limitation of the study.

Our response: Both men and women were included in the study. We have adjusted the column descriptor to make it clearer that the data presented refer to the number and percentage of men in the Non-PIGD and PIGD groups.

Comments: Also, were the participants excluded if they had a history of chronic illness? (diabetes, arthritis, hx of stroke etc?).

Our response: In addition to excluding participants who were “an inability to walk unassisted without a walking aid during the tests, a Mini-mental State Examination (MMSE) [17] score < 24, an atypical Parkinsonian syndrome evidence of psychosis, neuroleptic use, vertigo, epilepsy, stroke, transient ischaemic attacks, syncope, uncompensated heart failure or valvular heart disease”. We have added this information to section 2.1.

Reviewer 3 Report

This prospective cohort study of makes a well-founded study on falls in two Parkinson’s disease subtypes: Postural Instability  and Gait Difficulty (PIGD subtype) and manifesting tremor as main symptoms (Non-PIGD subtype). The paper concludes that compared with Non-PIGD, PIGD people were significantly more likely to suffer falls due to freezing of gait, balance-related falls and falls at home.

Even though some limitations of the study are described in lines 231-236, from a methodologically point of view an autonomous limitation section would be more than welcome. Authors clearly recognize the small sample of the study and the fact that fall location information was obtained for approximately only half the reported falls. The impact of this second limitation is well explained but I am missing futher arguments in order to justify the number of participants as a representative sample to reach scientific conclusions. Moreover, participants were recruited from Sydney, Australia. One may wonder if the election of this location provokes some kind of impact or bias in order to get universal findings. Please, go further in justifying the final sample and explaining other possible limitations with their impact on final results.

Concerning the conclusion I find it too short. Just there lines. Authors just mention that the paper´s findings document “the extent  to which people with PD with the PIGD subtype are at increased risk of falls, the circumstances in  which they fall and their disease-related, clinical and functional impairments”. In this regard, I am missing as a previous conclusion the fact that the PIGD people were significantly more likely to suffer falls than the non-PIGD one.

Any case, even though it is difficult to reach general conclusions taking into account the small sample and the local scope of the study, the paper is helpful for further developments.

Author Response

Comments: This prospective cohort study of makes a well-founded study on falls in two Parkinson’s disease subtypes: Postural Instability and Gait Difficulty (PIGD subtype) and manifesting tremor as main symptoms (Non-PIGD subtype). The paper concludes that compared with Non-PIGD, PIGD people were significantly more likely to suffer falls due to freezing of gait, balance-related falls and falls at home.

Even though some limitations of the study are described in lines 231-236, from a methodologically point of view an autonomous limitation section would be more than welcome. Authors clearly recognize the small sample of the study and the fact that fall location information was obtained for approximately only half the reported falls. The impact of this second limitation is well explained but I am missing further arguments in order to justify the number of participants as a representative sample to reach scientific conclusions. Moreover, participants were recruited from Sydney, Australia. One may wonder if the election of this location provokes some kind of impact or bias in order to get universal findings. Please, go further in justifying the final sample and explaining other possible limitations with their impact on final results.

Our Response: We have now included a discrete strengths and limitations paragraph in the discussion. This discusses the convenience nature and size of the sample, the possibility of some misclassifications of the fall types reported, the non-recording of some fall type information, and the limitations of sampling from a single site. We have also removed the statement that fall location data are likely representative of most falls suffered by people with the Non-PIGD and PIGD subtypes.

With regards to sample size, the current paper presents a secondary analysis from data that were collected from a sample of 113 people with PD (Latt et al., Mov Disord, 2009). The sample size is therefore based on power analyses computed to fit the purpose of the initial study (prospective study of falls for which 100 participants were considered a sufficient number to detect statistically significant differences in physiological, cognitive and clinical outcome measures between fallers and non-fallers (Latt et al., Mov Disord, 2009). We have therefore clarified in the participants sub-section 2.1 of the methods that: “this study comprised a secondary analysis from a prospective study of falls (Latt et al., Mov Disord, 2009)”. We have also included the sources also in the sub-section 2.1 from which participants were recruited. i.e. “Volunteers were recruited from the following sources: 36 (31%) from a hospital outpatient clinic, 15 (13%) from a volunteer database, 20 (18%) from PD support groups, and 42 (37%) from the general community”.

Comments: Concerning the conclusion I find it too short. Just there lines. Authors just mention that the paper´s findings document “the extent  to which people with PD with the PIGD subtype are at increased risk of falls, the circumstances in  which they fall and their disease-related, clinical and functional impairments”. In this regard, I am missing as a previous conclusion the fact that the PIGD people were significantly more likely to suffer falls than the non-PIGD one.

Our Response: We added the following sentences to section 5. Conclusions “Compared with Non-PIGD participants, PIGD participants were significantly more likely to suffer falls more falls overall as well as more falls due to freezing of gait, balance-related falls and falls at home. The PIGD group also performed significantly worse in a range of fall-related clinical and functional measures including general cognitive status, executive function, lower limb muscle strength, postural sway and the timed up and go test.”

Reviewer 4 Report

This manuscript describes a comparison between individuals with PD with more or less PIGD. These groups are compared on the number of falls, and a set of clinical/physical measures that were previously found to predict falls in other cohorts. This comparison showed that PD classified as PIGD-dominant motor subtype have more falls in the following year compared to a non-PIGD group. And this group scores worse on a set of clinical/physical measures.

I have a couple of general concerns:

I have trouble following the authors with the rationale, goal and importance of this study. The stated aims are to describe differences between motor subtypes of PD related to fall incidence, location and reason for falls and potential risk factors for falls. What do we gain from such a comparison? The number falls is very likely higher in the PIGD-group, and the current study shows that, this time in a prospective manner. Comparing the clinical and physical outcomes between groups shows that PIGD have troubles on most measures, which is expected based on the higher disease severity. The study, however, does not analyze if these measures were actual risk factors for falls in this (sub)sample, and if risk factors differed between PIGD and non-PIGD fallers. Parameters that differ between PIGD and non-PIGD are not necessarily factors that increase risk for falls.

In the conclusion the authors state that the clinical importance is that risk factors for falling have been identified, and that these can be targets for intervention depending on motor subtype. Was this the goal of the study? I recommend to adjust the introduction to provide a clear rationale and aim. Nonetheless, the conclusion is not substantiated by the data of the study.

Then there is an issue with the concept that is captured by PIGD-dominance (Nutt Mov Disord 2016 & Kotagal Ann Clin TRansl Neurol 2016. Is it disease stage, or do PIGD and TD represent distinct separate entities? Evidence has been building up against this idea. This makes it even more difficult to understand the goal of this study: why is the comparison interesting? And how does it help clinical practice?

The authors mention this concern in the discussion, and state that by adding disease duration to the statistical models to adjust for the difference in disease stage between the PIGD and non-PIGD groups. Still, the data may as well be interpreted as comparing two groups in different disease stages: data in Table 1 and 2 do support that.  

Disease duration is reported by the patient, recalling onset of first symptoms that is likely more easily recognized if the first symptom is tremor in contrast to axial symptoms, biasing the results. What happens if you use H&Y stage to adjust for disease severity, of total UPDRS score?

Please provide the coefficients of the adjusted models for all variables that were included in the models.

Why was fall rate not analysed in a model with disease duration/stage entered?

The authors should provide n for each H&Y stage rather than means (Table 3; H&Y is a ordinal scale).

Other concerns:

Methods:
- Was this study preregistered in any form? Or was is part of a larger study that was preregistered?

Why did the authors choose for the UPDRS instead of the MDS-UPDRS score?

Variability of step times is dependent on the number of steps. Please provide the average number of steps that were analyzed, and can you reflect if this exceeded the required n of steps (typically 20-30).
Results:

Table 1: Please provide the information on tremor scores and PIGD scores for each group

Table 1: What do the variables “Rigidity”, “Stooped posture”, … represent? Is this the n above a certain cut off value?

Discussion:

FOG is a main factor for falls in PD. And was highly prevalent in the PIGD group. Moreover, 23/37 fallers fell as a consequence of FOG. Would it not be more valuable to classify FOG rather than PIGD?

Author Response

Comments: This manuscript describes a comparison between individuals with PD with more or less PIGD. These groups are compared on the number of falls, and a set of clinical/physical measures that were previously found to predict falls in other cohorts. This comparison showed that PD classified as PIGD-dominant motor subtype have more falls in the following year compared to a non-PIGD group. And this group scores worse on a set of clinical/physical measures.

I have a couple of general concerns:

I have trouble following the authors with the rationale, goal and importance of this study. The stated aims are to describe differences between motor subtypes of PD related to fall incidence, location and reason for falls and potential risk factors for falls. What do we gain from such a comparison? The number falls is very likely higher in the PIGD-group, and the current study shows that, this time in a prospective manner. Comparing the clinical and physical outcomes between groups shows that PIGD have troubles on most measures, which is expected based on the higher disease severity. The study, however, does not analyze if these measures were actual risk factors for falls in this (sub) sample, and if risk factors differed between PIGD and non-PIGD fallers. Parameters that differ between PIGD and non-PIGD are not necessarily factors that increase risk for falls.

Our Response: We appreciate the reviewer’s point, but feel our study sample is too small to address fall risk factors within the two PD subtypes. In our revised paper we have been careful to ensure our discussion does not go beyond our study aims which are aimed at comparing the two PD subtypes on a range of clinical, physiological and cognitive factors – an approach we feel provides some indications of potential mechanisms for falls, particularly in the PD subtype. For example, we have replaced in the last paragraph of the section 4. Discussion, the following sentence “Our findings have important clinical implications in that they identify differential risk factors for falls that may be amenable to intervention according to PD motor subtypes” with “Our findings have important clinical implications in that they document clinical, medical and sensorimotor impairments in people with the PD PIGD subtype, some of which may be amenable to intervention”. We also know acknowledge that investigation of risk factors for falls within each subtype could be investigated in future research in the discussion (page 8).

Comments: In the conclusion the authors state that the clinical importance is that risk factors for falling have been identified, and that these can be targets for intervention depending on motor subtype. Was this the goal of the study? I recommend to adjust the introduction to provide a clear rationale and aim. Nonetheless, the conclusion is not substantiated by the data of the study.

Our Response: Thank you for pointing the inconsistency in our reporting of the study findings. As reported in our response to the previous comment, we have now rephrased the sentence starting with “Our findings have important clinical implications in that they document clinical, medical and sensorimotor impairments in people with the PD PIGD subtype, some of which may be amenable to intervention”. As stated in the end of the introduction, the goal of the study was to identify the difference between subtypes in incidence of falls, falls locations and type of falls and in a range of disease-related, cognitive, functional and mobility measures. We have modified part of the conclusion to be in line with the aims of this study. We feel that the conclusions reproduced below are now in line with the study aims.

“The study findings document the extent to which people with PD with the PIGD subtype are at increased risk of falls, the circumstances in which they fall and their disease-related, clinical and functional impairments. Compared with Non-PIGD participants, PIGD participants were significantly more likely to suffer falls more falls overall as well as more falls due to freezing of gait, balance-related falls and falls at home. The PIGD group also performed significantly worse in a range of fall-related clinical and functional measures including general cognitive status, executive function, lower limb muscle strength, postural sway and the timed up and go test. This information may prove useful for informing cognitive, physical and environmental interventions to prevent falls in this high risk group.”

Comments: Then there is an issue with the concept that is captured by PIGD-dominance (Nutt Mov Disord 2016 & Kotagal Ann Clin TRansl Neurol 2016. Is it disease stage, or do PIGD and TD represent distinct separate entities? Evidence has been building up against this idea. This makes it even more difficult to understand the goal of this study: why is the comparison interesting? And how does it help clinical practice?

Our Response: We do acknowledge in the discussion that the TD-PIGD subtype distinction may reflect different stages of Parkinson's disease rather than different disorders. However, we feel the study has important implications in that the PIGD group also performed significantly worse in a range of fall-related clinical and functional measures including general cognitive status, executive function, lower limb muscle strength, postural sway and the timed up and go test – factors that go beyond PD severity. As indicated in our discussion we hope this information may prove useful for informing cognitive, physical and environmental interventions to prevent falls in this high risk group.

Comments: The authors mention this concern in the discussion, and state that by adding disease duration to the statistical models to adjust for the difference in disease stage between the PIGD and non-PIGD groups. Still, the data may as well be interpreted as comparing two groups in different disease stages: data in Table 1 and 2 do support that.  Disease duration is reported by the patient, recalling onset of first symptoms that is likely more easily recognized if the first symptom is tremor in contrast to axial symptoms, biasing the results. What happens if you use H&Y stage to adjust for disease severity, or total UPDRS score?

Our Response: We have carefully considered this alternative analysis, but feel it is problematic as the PIGD/ non-PIGD classification, UPDRS total score and Hoehn and Yahr stage measures comprise share components and are therefore not independent of each other. To circumvent inevitable over-adjustment we adjusted for disease duration instead - as this is an independent measure. We hope our discussion of this issue is adequate.

“It has been suggested the TD-PIGD subtype distinction may reflect different stages of Parkinson's disease rather than different disorders [29], and as anticipated, the PIGD group had longer disease durations than the Non-PIGD group in our cohort. Accordingly, we adjusted for PD duration in our subtype comparisons and some measures (motor complications assessed by UPDRS part IV, dyskinesia, anticholinergic medication, risk of falls assessed by the PPA and controlled leaning balance) did not remain statistically significant. However, many disease-related, cognitive, sensorimotor, balance, gait and mobility measures did remain significant discriminators of PD subtype after adjustment for disease duration. Therefore, it appears the PD subtype classification, as used in this study, is useful for identifying people with PD at increased fall risk as well as elucidating possible underlying causes of falls.”

Comments:Please provide the coefficients of the adjusted models for all variables that were included in the models.

Our Response: We are not sure if we understand this question and how it relates to our negative binomial regression models.

Comments: Why was fall rate not analysed in a model with disease duration/stage entered?

Our Response: We have included adjusted analyses for disease duration in the revised manuscript. Updated results are present in Table 3.

Comments: The authors should provide n for each H&Y stage rather than means (Table 1; H&Y is a ordinal scale).

Our Response: Hoehn and Yahr stages are now included in Table 1.

Comments: Other concerns: Methods: Was this study preregistered in any form? Or was is part of a larger study that was preregistered?

Our Response: This study comprises a secondary analysis from the data presented in Latt et al., 2009 and was not pre-registered. We have now indicated this in the first sentence of the methods section (page 2) that this study comprises a secondary analysis from a prospective study of falls (Latt et al., Mov Disord, 2009).

Comments: Why did the authors choose for the UPDRS instead of the MDS-UPDRS score?

Our Response: The UPDRS was used as this was the assessment recommended by the Movement Disorders Society at the time the data were collected (2007).

Comments:Variability of step times is dependent on the number of steps. Please provide the average number of steps that were analysed and can you reflect if this exceeded the required n of steps (typically 20-30).

Our Response: The average of steps performed was 32 (range 14-204). These data are now included in the revised manuscript.

Comments: Results: Table 1: Please provide the information on tremor scores and PIGD scores for each group

Our Response: The information regarding TD and PIGD scores has been added to Table 1.

Comments:Table 1: What do the variables “Rigidity”, “Stooped posture”, … represent? Is this the n above a certain cut off value?

Our Response: A table legend has now been added which indicates the entries for the rigidity, axial posture, bradykinesia and dyskinesia variables relate to their presence being noted in the UPDRS assessment. We have also clarified our definitions in sub-section 2.2.2. PD-related and health measures “Presence of Rigidity, Axial Posture, Bradykinesia and Dyskinesia according to the UPDRS items”

Comments: Discussion: FOG is a main factor for falls in PD. And was highly prevalent in the PIGD group. Moreover, 23/37 fallers fell as a consequence of FOG. Would it not be more valuable to classify FOG rather than PIGD?

Our Response: The results do indicate FOG is a major contributor to falls in the PIGD group and we agree contrasting those with and without FOG would be of interest. However, we feel that the current comparison is also of value, and would therefore prefer to retain this paper in its current form.

Comments: References used in this response letter.

Our Response: Latt, M.D.; Lord, S.R.; Morris, J.G.L.; Fung, V.S.C. Clinical and physiological assessments for elucidating falls risk in Parkinson’s disease. Mov Disord 2009, 24, 1280-1289.